# The Gut Microbiome and Epigenomic Reprogramming: Mechanisms, Interactions, and Implications for Human Health and Disease

**DOI:** 10.3390/ijms26178658

**Published:** 2025-09-05

**Authors:** Noelle C. Rubas, Amada Torres, Alika K. Maunakea

**Affiliations:** 1Department of Molecular Bioscience and Bioengineering, University of Hawai‘i at Mānoa, Honolulu, HI 96822, USA; 2Department of Anatomy, Biochemistry, and Physiology, John A. Burns School of Medicine, University of Hawai‘i at Mānoa, Honolulu, HI 96822, USA

**Keywords:** gut microbiome, epigenetic regulation, DNA methylation, short-chain fatty acids, host–microbe interactions, precision medicine

## Abstract

The human gut microbiome is a metabolically active and ecologically dynamic consortium that profoundly influences host physiology, in part by modulating epigenetic mechanisms such as DNA and RNA methylation. These modifications regulate gene expression and phenotypic plasticity and are shaped by a combination of environmental factors, such as diet, stress, xenobiotics, and bioactive microbial metabolites. Despite growing evidence linking microbial signals to host epigenetic reprogramming, the underlying molecular pathways remain incompletely understood. This review highlights recent mechanistic discoveries and conceptual advances in understanding microbiome–host epigenome interactions. We discuss evolutionarily conserved pathways through which gut microbiota regulate host methylation patterns, including one-carbon metabolism, polyamine biosynthesis, short-chain fatty acid signaling, and extracellular vesicle-mediated communication. We also examine how host factors such as aging, diet, immune activity, and sociocultural context reciprocally influence microbial composition and function. Beyond basic mechanisms, we outline translational frontiers—including biomarker discovery, live biotherapeutic interventions, fecal microbiota transplantation, and adaptive clinical trial designs—that may enable microbiome-informed approaches to disease prevention and treatment. Advances in high-throughput methylation mapping, artificial intelligence, and single-cell multi-omics are accelerating our ability to model these complex interactions at high resolution. Finally, we emphasize the importance of rigorous standardization and ethical data governance through frameworks such as the FAIR and CARE principles. Deepening our understanding of how the gut microbiome modulates host epigenetic programs offers novel opportunities for precision health strategies and equitable clinical translation.

## 1. Epigenetics at the Host–Microbiota Interface

All multicellular organisms have co-evolved with diverse communities of microorganisms that are vital for host physiology. Host-microbiome interactions have shaped animal evolution and diversification, particularly by enabling nutrient breakdown and absorption [1,2]. To coordinate these interactions, most organisms have developed sophisticated epigenetic mechanisms that reversibly regulate gene expression in response to environmental cues, thereby supporting phenotypic plasticity and adaptation. Epigenetics broadly refers to heritable phenotypic changes that occur without alterations to the DNA sequence. Major classes of epigenetic mechanisms include DNA and RNA methylation [3], histone modifications driving chromatin remodeling [4], and the actions of non-coding RNAs [5]. Although distinct, these processes often function together, adding layers of regulatory complexity that influence gene expression, genome stability, development, and cellular differentiation, function, and identity.

The gut microbiome—comprising bacteria, viruses, fungi, and other microorganisms inhabiting the human gastrointestinal tract (GI)—is most densely concentrated in the colon, particularly the cecum, where microbial communities replicate and interact with the host [6,7]. This ecosystem contributes to numerous physiological functions, including nutrient release, vitamin production, digestion, energy balance, and regulation of carbohydrate and lipid metabolism. Anaerobic fermentation produces short-chain fatty acids (SCFAs) and other metabolites that sustain intestinal and systemic health. In addition, the gut microbiome primes and modulates host immune and neuroimmune responses [8].

In return, the host provides a stable environment and dietary substrates that sustain microbial metabolism, fostering a dynamic and mutually beneficial relationship. Quorum-sensing molecules and other bioactive compounds mediate this bidirectional crosstalk, shaping host defense and preventing pathogen overgrowth [9,10]. Despite extensive research underscoring the microbiome’s role in health, the precise mechanisms by which microorganisms modulate host epigenetic landscapes remain incompletely defined and require further study.

To date, accumulating evidence supports a bidirectional relationship: host genetics and epigenetic states can shape the composition and function of the microbiota, while microbial communities, in turn, influence host epigenetic regulation through diet- and environment-dependent mechanisms (Figure 1). This reciprocity highlights the gut microbiome as both a target and driver of molecular adaptation. By modulating chromatin accessibility and transcriptional programs, microbial metabolites and signaling molecules may reprogram host cell states in ways that affect immunity, metabolism, neurobiology, and disease susceptibility. Conversely, epigenetic regulation in host tissues can impose selective pressures that favor particular microbial taxa, suggesting that the microbiome is an active participant in evolutionary and developmental trajectories rather than a passive bystander.

Clarifying this interplay carries profound implications. Understanding how microbial factors reshape host epigenetic landscapes could reveal new therapeutic avenues for alleviating genetic and epigenetic constraints on health, while also informing microbiome-targeted strategies such as dietary interventions, probiotics, and metabolite-based therapeutics. At the same time, dissecting how host epigenetic states influence microbial ecology may uncover principles for designing precision approaches to maintain or restore microbial balance in chronic disease.

Herein, we review emerging evidence of how the gut microbiome modulates host epigenetic processes, with particular emphasis on DNA and RNA methylation. We highlight key cellular pathways and mediator molecules through which this regulation occurs, and we discuss future strategies to disentangle these complex, reciprocal interactions in health and disease. Framed within the era of precision medicine, these insights underscore the potential of microbiome–epigenome crosstalk as a foundation for next-generation diagnostics and therapeutics.

## 2. Evolutionary Origins and Comparative Mechanisms of Methylation

Understanding the evolution of epigenetic regulation is key to contextualizing how humans respond to microbial signals at the molecular level. While this review centers on the human gut microbiome and its impact on host methylation, insights from yeast, plants, and model animals reveal conserved enzymatic pathways, distinct DNA and RNA methylation patterns, and adaptive strategies that underlie epigenomic plasticity. Such comparative perspectives enhance our interpretation of microbiome–epigenome interactions and guide the development of cross-kingdom models and bioengineered systems.

### 2.1. Overview of DNA and RNA Methylation Across Domains

Epigenetic regulation is conserved across most domains of life, with DNA and RNA methylation systems essential for genome defense, transcriptional control, and developmental plasticity. DNA methylation involves the covalent addition of a methyl group to the fifth carbon of the cytosine ring, generating 5-methylcytosine (5mC). In vertebrates, this modification occurs predominantly at 5′-CG-3′ (CpG) dinucleotides, while in other taxa it can also appear in non-CpG contexts, though to varying degrees. RNA modifications such as N6-methyladenosine (m6A) have likewise emerged as key regulators of gene expression, influencing RNA stability, splicing, and translation. These marks are dynamically controlled by writer, reader, and eraser proteins, many of which are evolutionarily conserved and act concertedly in a context-dependent manner [11,12,13,14,15,16,17,18,19] (Table 1).

### 2.2. DNA Methylation and Small RNA Regulatory Functions in Microbiota

In bacteria, post-replicative DNA methylation commonly involves three modifications: 5-methylcytosine (5mC), N6-adenine methylation (6mA), and N4-methylcytosine (4mC) [20,21]. Most research has focused on 5mC, which functions within restriction-modification (RM) systems that protect bacterial genomes from foreign DNA. These RM systems, comprising endonucleases and methyltransferases, are classified into four main types (I–IV) based on their genetic organization, biochemical properties, and recognition mechanisms.

Bacteria have become valuable models for studying epigenetics, providing tools such as isoschizomer restriction enzymes (e.g., *Msp*I and *Hpa*II) for surveying genome-wide DNA methylation states [38,39]. Bacterial DNA methylation influences clinically relevant traits, including host colonization, biofilm formation, sporulation, and virulence [40]. In addition, bacterial small RNAs (sRNAs)—typically 50 to 500 nucleotides in length—act as versatile regulators by base-pairing with target mRNAs to modulate gene expression [41,42,43], adding another layer of regulatory complexity. For example, certain RNA methyltransferases uniquely modify adenosine residues at specific carbon positions. Mutations disrupting the RNA methyltransferase *Ksg*A have been linked to antibiotic resistance in *Escherichia coli*, highlighting the physiological importance of RNA methylation in bacterial adaptation [22].

### 2.3. Yeast and Fungal DNA Methylation

Historically, yeast such as *Saccharomyces cerevisiae* were considered to lack DNA methylation [44]; however, low levels of 5mC and noncanonical methylation systems have been reported in some fungal species, including *Candida albicans* and *Neurospora crassa*. In *N. crassa*, DNA methylation occurs via DIM-2 methyltransferase and plays a role in silencing transposable elements [45]. RNA-based mechanisms such as small interfering RNAs (siRNAs) and heterochromatic silencing also contribute to epigenetic control in fungi, providing a functional parallel to mammalian systems.

Fungal DNA methylation plays diverse biological roles, including regulation of reproductive development, phase transitions, and phenotypic variation. In the human gut, common fungal genera include *Aspergillus*, *Candida*, *Debaryomyces*, *Penicillium*, *Pichia*, and *Saccharomyces* [45]. Among these, *Candida albicans* exhibits DNA methylation associated with key processes such as its dimorphic switch and iron metabolism [24]. Fungi encode distinct DNA methyltransferases (DNMTs), such as Masc1, which mediates methylation-induced premeiotic mutations, and DNMT5, which targets repetitive elements. Interestingly, some model yeasts like *Saccharomyces cerevisiae* and *Schizosaccharomyces pombe* display little to no detectable DNA methylation. However, RNA-directed DNA methylation (RdDM)-like pathways [30], functionally analogous to those in plants (described below), have been identified in members of the Basidiomycota phylum [24], suggesting conserved epigenetic regulatory mechanisms across fungal lineages.

### 2.4. Plant DNA Methylation

Plants possess a highly structured and complex DNA methylation system that plays critical roles in development, transposon silencing, and responses to environmental stress. Cytosine methylation in plants occurs not only in the CG context, but also in CHG and CHH contexts (where H = A, T, or C), each regulated by distinct enzymatic pathways. CG methylation is maintained by DNA methyltransferase 1 (MET1), while chromomethylases such as CMT2 and CMT3 maintain methylation at CHG sites. CHH methylation, which is more dynamic, is established *de novo* by DRM2 or CMT2 [46]. Demethylation in plants is an active process involving DNA glycosylases, including Repressor of Silencing 1 (ROS1) and members of the demeter-like family [27,28,29]. One of the most distinctive features of plant epigenetics is RdDM, a potent gene silencing mechanism that lacks a direct mammalian counterpart but offers a conceptual framework for understanding small RNA-mediated chromatin regulation. In this process, RNA polymerase IV synthesizes single-stranded RNAs, which are converted into double-stranded RNAs and processed into siRNAs [43]. Concurrently, RNA polymerase V transcribes non-coding RNAs [47] that guide the siRNA–Argonaute complex [48,49] to specific genomic loci, recruiting *de novo* methyltransferases such as DRM2 [47,50,51] to deposit new methylation marks and reinforce silencing.

### 2.5. DNA Methylation and RNA Regulatory Activity in Animals

In animals, DNA methylation is primarily established by DNMT3A/3B and maintained by DNMT1. These marks are vital for X-inactivation, genomic imprinting, and transposon suppression. Dynamic demethylation is facilitated by the TET family of enzymes, which oxidize 5mC to 5hmC and further intermediates [52]. RNA methylation, particularly m6A, is recognized by YTH domain-containing proteins and has emerged as a major layer of post-transcriptional gene regulation [53,54,55]. These pathways are responsive to environmental cues and metabolites—including those derived from the microbiota—making them central to the microbiome–host epigenome crosstalk.

Cytosine methylation at the position 5 is the predominant modification in most animal genomes [52], although 6mA has been reported in some lineages [25,55]. Comparative methylome analyses show that while most animal genomes contain DNA methylation, certain animal model organisms such as *Drosophila melanogaster* and *Caenorhabditis elegans* exhibit minimal to non-detectable DNA methylation [24]. In humans and rodents, DNA methylation is a dynamic, tissue-specific process influenced by age, environment, lifestyle [56,57,58], and disease status [59]. Indeed, we observed evolutionary conservation of tissue-specific DNA methylation patterns as a common feature in regulating cellular plasticity via alternative promoter usage and RNA splicing [57,58]. The human genome encodes three principal DNA methyltransferases (DNMTs): DNMT1, which maintains existing methylation patterns during DNA replication, and DNMT3A and DNMT3B, which catalyze *de novo* methylation. In contrast, active DNA demethylation is mediated by the ten-eleven translocation (TET) family of dioxygenases (TET1–TET3), which sequentially oxidizes 5mC to generate 5hmC and other oxidized derivatives.

In parallel, RNA methylation—most notably m6A—represents one of the most abundant post-transcriptional RNA modifications. It regulates multiple aspects of RNA metabolism, including stability, alternative splicing, translation efficiency, and RNA–protein interactions, and plays essential roles in development and disease [3,32,37,54]. This dynamic mark is regulated by “writers” (e.g., METTL3, METTL14, WTAP), “erasers” (e.g., FTO, ALKBH5), and “readers” (e.g., YTH domain protein and IGF2BP family) [33,35]. Methylation affects diverse RNA species other than mRNA, including circular RNAs, tRNAs, rRNAs, and multiple classes of small RNAs (miRNAs, siRNAs, piRNAs, snoRNAs, tRFs) [60]. The case of DNMT2, the most highly conserved methyltransferase across eukaryotes, underscores the evolutionary flexibility of epigenetic enzymes. Although originally classified as a DNA methyltransferase, DNMT2 primarily catalyzes cytosine-5 methylation of tRNAs, with a strong preference for tRNA^Asp^; remarkably, DNMT2 from one species can functionally complement *Dnmt2*-deficient models in others, restoring tRNA^Asp^ methylation in mice, *Arabidopsis*, and *Drosophila* [36,37]. Collectively, these modifications shape cellular regulatory networks and may also underlie evolutionarily conserved host–microbiome interactions.

## 3. Mechanistic Insights into Microbiome-Induced Host Epigenetic Regulation

The gut microbiome plays an increasingly recognized role in modulating the host epigenome by influencing DNA and RNA methylation though diverse mechanisms outlined in Table 2. Herein, we summarize key pathways through which microorganisms impact host methylation processes, focusing on microbial presence, one-carbon metabolism, fermentation products, extracellular vesicles (EVs), and inflammatory modulation within intestinal epithelial cells (IECs).

### 3.1. Microbial Presence Shapes Host DNA and RNA Methylation Profiles

The presence or absence of gut microbiota can profoundly influence the expression and activity of the host’s methylation machinery. Comparative studies between germ-free (GF) mice and conventionally raised (CR) or specific pathogen-free (SPF) mice have demonstrated that microbial colonization significantly reshapes the host methylome.

For instance, GF mice exhibit disorganized lymphoid architecture and impaired immune responses, in contrast to SPF mice with normal microbiota colonization [72]. In one study, Reduced Representation Bisulfite Sequencing (RRBS) combined with transcriptomic analysis of IECs identified over 100 genomic regions with microbiota-dependent differences in DNA methylation and gene expression. Notably, microbial presence influenced the expression of key epigenetic regulators, including DNMT3A and TET3 [73]. Another investigation revealed that CR mice displayed lower levels of global promoter DNA methylation in IECs compared to GF mice, alongside enhanced regulatory activity and elevated TET3 expression. Genetic knockout of TET2 and TET3 in this context led to increased methylation and decreased expression of genes located in hypermethylated regions. These epigenetic changes were accompanied by the enrichment of histone-modifying regulatory elements, suggesting a direct mechanistic connection between microbial colonization and host chromatin remodeling [74]. Together, these studies underscore the critical role of gut microbiota in shaping the host epigenetic landscape, particularly during immune system maturation and intestinal development.

Emerging evidence also indicates that the gut microbiome may influence RNA modifications, particularly m6A, a key epitranscriptomic mark that regulates RNA stability, splicing, and translation. Liquid chromatography–mass spectrometry (LC-MS) analyses have revealed distinct m6A methylation profiles in multiple organs—including the brain, liver, and intestines—of SPF mice compared to their GF counterparts [75]. These findings suggest that microbial colonization exerts systemic effects on RNA methylation across diverse tissues. Specifically, certain gut microbes such as *Akkermansia muciniphila* and *Lactiplantibacillus plantarum* have been shown to modulate host m6A marks in genes involved in cell growth, differentiation, and metabolic regulation [76].

These microbial influences likely occur through signaling pathways and metabolite-mediated mechanisms that regulate the expression or activity of m6A “writers,” “erasers,” and “readers”—including METTL3/14 (methyltransferases), FTO/ALKBH5 (demethylases), and YTH-domain proteins. Although still an emerging area of study, these findings point to an underexplored layer of host–microbiome interaction that operates at the level of post-transcriptional gene regulation.

### 3.2. Role of Microbial One-Carbon Metabolism

One-carbon metabolism serves as a critical biochemical nexus linking nutritional status, microbial activity, and epigenetic regulation. This pathway generates S-adenosylmethionine (SAM), the universal methyl donor required for DNA, RNA, and histone methylation processes [62]. Central to one-carbon metabolism are the folate cycle, methionine cycle, and trans-sulfuration pathway, all of which depend on essential micronutrients—namely folate (vitamin B9), riboflavin (vitamin B2), pyridoxine (vitamin B6), and cobalamin (vitamin B12) [77,78,79]. Because humans cannot synthesize these cofactors endogenously, they must obtain them through dietary intake and gut microbial biosynthesis [80].

Gut-resident microorganisms—particularly taxa within the Bacteroidetes, Fusobacteria, and Proteobacteria phyla—play a key role in producing or metabolizing these vitamins [61], thereby modulating the host’s methyl group bioavailability. Dysbiosis or shifts in microbial composition can impair this balance. For example, an overrepresentation of choline-degrading bacteria may simulate dietary choline deficiency, reducing SAM levels and contributing to global hypomethylation and associated metabolic dysfunctions [81]. Conversely, Lactobacillus [62] and Bifidobacterium [63] species are known to synthesize folate and contribute to maintaining epigenetic homeostasis by supporting adequate methyl donor supply.

### 3.3. Intersection of One-Carbon and Polyamine Metabolism

Polyamines (PAs)—including putrescine, spermidine, and spermine—are small, positively charged molecules that play essential roles in cell proliferation, gene regulation, stress responses, and epigenetic maintenance [82]. These bioactive compounds are acquired through dietary intake and are also synthesized *de novo* by both host cells and the gut microbiota [83]. Within the lower GI [84], specific microbial taxa, including *Lactobacillus* spp. [64] and *Shewanella xiamenensis* [65], convert dietary arginine into ornithine, and subsequently into putrescine via ornithine decarboxylase (ODC) activity. Additional enzymatic steps then generate spermidine and spermine, utilizing aminopropyl groups derived from SAM.

Importantly, polyamine metabolism is tightly interlinked with one-carbon metabolism, as it consumes and regulates levels of SAM and its decarboxylated form (dcSAM). For example, elevated intracellular concentrations of spermine can inhibit both ODC and adenosylmethionine decarboxylase (AdoMetDC), resulting in increased SAM availability and consequently enhanced DNMT activity [85]. Through this regulatory feedback, polyamines indirectly modulate DNA methylation patterns, contribute to genomic stability, and influence immune homeostasis and inflammatory responses.

### 3.4. Impact of Microbial Fermentation Products

Microbial fermentation of dietary carbohydrates in the colon leads to the production of SCFAs—primarily acetate, propionate, and butyrate—which serve not only as energy sources but also as potent signaling molecules within the intestinal epithelium. Among their diverse biological effects, SCFAs are best known for their role as histone deacetylase (HDAC) inhibitors, thereby promoting a more open chromatin configuration and enhancing gene expression [66,68]. In addition to histone modifications, SCFAs can indirectly influence DNA methylation by modulating substrate availability for methylation pathways and altering the expression of DNMTs. Notably, in GF mouse models, SCFA supplementation can partially restore methylation and chromatin modification patterns that are otherwise dependent on the presence of gut microbiota [79].

Beyond SCFAs, other microbiota-derived metabolites significantly impact epigenetic regulation. For example, intermediates of the tricarboxylic acid (TCA) cycle such as succinate and fumarate can act as competitive inhibitors of α-ketoglutarate-dependent dioxygenases, including TET enzymes, thereby suppressing active DNA and histone demethylation processes. Furthermore, certain dietary glucosinolates, when metabolized by gut bacteria like *Bacteroides thetaiotaomicron* [69], are converted into bioactive isothiocyanates (e.g., sulforaphane), which have been shown to inhibit DNMT activity and alter host DNA methylation patterns [70]. Collectively, these microbial metabolites serve as epigenetic modulators, linking diet, microbial metabolism, and host gene regulation through both direct enzymatic inhibition and substrate competition in chromatin remodeling pathways.

### 3.5. Extracellular Vesicles as Epigenetic Messengers

Microbial EVs—especially outer membrane vesicles (OMVs) secreted by Gram-negative bacteria—are increasingly recognized as potent mediators of inter-kingdom communication. These nanoscale vesicles encapsulate nucleic acids, proteins, metabolites, and lipids, and can traverse epithelial barriers to deliver their cargo into host cells, including epithelial, immune, and stromal tissues [86,87]. Once internalized, microbial EVs have the capacity to modulate host signaling cascades, epigenetic machinery, and gene expression profiles at local and distant sites.

A striking example of such interaction is provided by *Pseudomonas aeruginosa*, whose OMVs were shown to induce widespread hypomethylation in human lung macrophages, thereby enhancing the accessibility of distal regulatory elements linked to immune defense genes [71]. These changes not only affect transcriptional outputs but may also prime immune cells for subsequent inflammatory or tolerogenic responses. Similar effects have been observed in intestinal systems, where OMVs derived from *Bacteroides fragilis* carry PSA and influence Foxp3 regulatory T cell expression, potentially through epigenetic remodeling of enhancer regions, further discussed below.

### 3.6. Epigenetic Modulation of the Inflammatory Response

The gut microbiome plays a critical role in maintaining intestinal homeostasis, in part by shaping the epigenetic responsiveness of IECs to microbial stimuli. One well-characterized example involves the epigenetic repression of Toll-like receptor 4 (TLR4)—a key pattern recognition receptor involved in detecting lipopolysaccharide (LPS) from Gram-negative bacteria. In IECs, TLR4 expression is attenuated through a combination of DNA hypermethylation and histone deacetylation at its promoter region. This process is mediated by the zinc finger protein ZNF160, which recruits the co-repressor KAP1 (KRAB-associated protein 1) to establish a repressive chromatin state [88]. As a result, IECs exhibit reduced sensitivity to LPS, thereby mitigating excessive inflammatory responses to the dense microbial populations in the gut lumen. This so called “epigenetic buffering” of microbial reactivity is crucial for mucosal tolerance, helping to prevent inappropriate immune activation while preserving host defense. It also underscores the tissue-specific tuning of host epigenetic programs in response to the local microbial environment.

Collectively, the interconnected pathways discussed—including nutrient provisioning for methylation reactions, production of bioactive metabolites (e.g., SCFAs, polyamines, isothiocyanates), extracellular vesicle-mediated communication, and immune receptor regulation—highlight the multifaceted ways in which the gut microbiota modulates host DNA and RNA methylation (Figure 1). Nevertheless, the precise molecular mechanisms by which microbial signals interface with host chromatin regulators—such as DNMTs, TET enzymes, and histone modifiers—remain incompletely defined and represent an urgent area for future research.

## 4. Host Epigenetic Landscape Shapes Microbiome Composition and Function

The relationship between the gut microbiome and the host epigenome is fundamentally bidirectional. While microbial communities exert significant influence on host DNA and RNA methylation, accumulating evidence indicates that the host’s genetic and epigenetic landscape actively shape the composition, diversity, and metabolic function of the microbiota [89,90]. This reciprocal regulation plays a central role in maintaining physiological homeostasis and responding to environmental stimuli, with important implications for precision medicine.

Recent studies suggest that host genomic variants, particularly in genes related to immunity [91,92], epithelial integrity, and metabolism [93], can influence microbial colonization patterns and niche selection. Additionally, host epigenetic modifications—including methylation of genes involved in mucosal barrier function, antimicrobial peptide expression, and glycosylation—can create a selective environment that favors the expansion or suppression of specific microbial taxa [94].

Beyond genetic and epigenetic determinants, extrinsic host factors such as diet, aging, and sociocultural context significantly modulate microbiota structure and function [95,96,97,98,99,100,101,102,103]. These influences interaction with host regulatory pathways to shape the dynamic ecosystem of the gut. This section summarizes emerging evidence on how host-intrinsic factors and environmentally responsive epigenetic programs contribute to shaping the microbiota, highlighting their role in reinforcing the feedback loop between host and microbial signals.

### 4.1. Host Genetic Determinants Influence Microbial Composition

Host genetic variation, particularly in genes governing immune function, plays a pivotal role in shaping the structure and diversity of the gut microbiota. For instance, mutations in the MEFV gene, which encodes the inflammasome-associated protein pyrin, are linked to familial Mediterranean fever (FMF) and have been shown to significantly alter microbial profiles. Affected individuals often exhibit reduced microbial diversity and distinct shifts in bacterial composition, which tend to correlate with disease flares and inflammatory status [96,97]. Such mutations are associated with reduced microbial diversity and compositional shifts that often parallel disease activity.

Twin studies further underscore the heritability of microbial traits, revealing that certain microbial taxa—such as members of the *Christensenellaceae* family—demonstrate high concordance among monozygotic twins, even when raised in different environments [95]. This genetic control is reinforced by genome-wide association studies (GWAS) that have identified single nucleotide polymorphisms (SNPs) associated with the relative abundance of key bacterial groups, including *Bifidobacterium*, *Lactobacillus*, and *Faecalibacterium*.

Additionally, in vivo evidence from animal gene knockout models lends further support: targeted deletion of host genes involved in immune signaling, epithelial barrier function, and metabolic regulation leads to reproducible changes in microbial community structure. Indeed, at least 30 distinct genomic loci have been implicated in regulating microbiota composition in such models [98], indicating a functional, genotype-dependent axis of host–microbiota interaction. Together, these findings highlight the importance of host genetics in establishing and maintaining a stable and health-promoting microbial ecosystem.

### 4.2. Nutritional Modulation of the Gut Microbiota

Diet is one of the most powerful and modifiable environmental factors influencing gut microbial diversity, abundance, and metabolic output. Dietary components not only determine nutrient availability for microbial fermentation but also modulate the production of epigenetically active metabolites, such as SCFAs, polyamines, and vitamins essential for one-carbon metabolism.

One widely studied dietary intervention is the low-FODMAP diet, which restricts fermentable oligosaccharides, disaccharides, monosaccharides, and polyols. This approach has proven to be clinically effective in managing irritable bowel syndrome (IBS) symptoms, in part by restructuring the gut microbiota [99]. Multi-center trials have consistently demonstrated that a low-FODMAP diet reduces populations of fermentative and SCFA-producing bacteria—such as certain *Clostridium* and *Bacteroides* species—while promoting the growth of beneficial taxa, including *Bifidobacterium longum* and *Anaerostipes propionicum* [100]. These microbiota shifts correlate with improved gastrointestinal function, reduced inflammation, and alleviation of IBS symptom severity.

Beyond gastrointestinal disorders, diet–microbiome–epigenome interactions are increasingly recognized as modulators of systemic immunity and neurodevelopment. For instance, ketogenic diet interventions in children with autism spectrum disorder (ASD) have been associated with favorable shifts in the gut microbiome including an increase in butyrate-producing bacteria, reduction in proinflammatory cytokines, and changes in the expression of microRNAs linked to brain-derived neurotrophic factor (BDNF) signaling [101]. These findings suggest a mechanistic link between diet-induced microbial changes and epigenetically regulated neural and immune pathways.

Together, these examples highlight the potential of dietary strategies to modulate the microbiota and, through it, the host epigenome, opening novel avenues for personalized nutrition and therapeutic interventions.

### 4.3. Aging, Psychosocial Stress, and the Microbiome–Epigenome Axis

Aging is accompanied by progressive changes in both the epigenome and the gut microbiome, which together shape immune function, metabolic health, and disease susceptibility. Epigenetic clocks—which estimate biological age based on DNA methylation patterns at specific CpG sites—have become valuable tools for assessing age-related physiological decline [56,92,102]. These clocks often reveal discrepancies between chronological and biological age, particularly in populations exposed to chronic stress or environmental insults. In studies involving Native Hawaiian and Pacific Islander (NHPI) cohorts, accelerated epigenetic aging was found to correlate with declines in beneficial taxa such as *Actinobacteria* and *Bacteroides*, and increases in proinflammatory taxa, including *Deferribacteres* and *Proteobacteria* [103]. These microbial shifts mirror age-related immune dysregulation and systemic inflammation, commonly referred to as inflammaging.

Beyond aging, mental health and psychosocial stress are increasingly recognized as modulators of the microbiome–epigenome axis. Individuals experiencing lower self-esteem and elevated psychological stress often exhibit increased circulating levels of proinflammatory cytokines such as IL-8 and TNF-α. These cytokine profiles are associated with distinct gut microbial configurations and differential DNA methylation patterns in peripheral immune cells [92]. Such epigenetic changes may mediate the biological imprint of chronic psychosocial stress, linking environmental adversity to immune dysfunction and microbial dysbiosis.

These effects are particularly pronounced in underserved populations such as NHPIs, where socioeconomic disparities contribute to both microbial imbalances and epigenetic alterations, compounding health risks across generations. Understanding these interconnected dynamics is crucial for designing equitable interventions that account for social determinants of health.

### 4.4. An Integrated View

Collectively, these findings underscore the dynamic and reciprocal nature of host–microbiome interactions. Host genetic architecture, dietary patterns, lifespan-related epigenetic remodeling, and the broader sociocultural environment play pivotal roles in shaping the composition, diversity, and functional capacity of the gut microbiome. In turn, microbial communities influence host physiology by modulating epigenetic regulators, including DNA and RNA methylation, histone modifications, and non-coding RNA networks that concertedly act to shape health trajectories.

This bidirectional feedback loop enables the epigenome to act as both a sensor and effector, translating environmental and microbial signals into heritable changes in gene expression. Through this framework, the host epigenome contributes to microbiota stability, adaptability, and resilience, which are essential for maintaining immune homeostasis, metabolic balance, and neurobehavioral integrity. Disruption of this delicate interplay—via genetic predisposition, dietary shifts, psychosocial stress, or aging—can predispose individuals to a wide spectrum of diseases, from inflammatory and metabolic disorders to neurodevelopmental and psychiatric conditions.

A systems-level understanding of these host–microbiome–epigenome interactions is critical for developing precision medicine strategies that account for individual variability and sociocultural context.

## 5. Dissecting Cellular Mechanisms of Microbiome–Epigenome Crosstalk: Mediator Molecules and Pathways

Although substantial progress has been made in linking the gut microbiome to host epigenetic regulation, the precise cellular signaling pathways and molecular mediators that orchestrate this dialog remain incompletely understood. Recent advances in multi-omics, gnotobiotic models, and chromatin profiling have begun to elucidate the complex interplay between microbial signals and host transcriptional regulators, offering promising avenues for therapeutic intervention. A particularly well-characterized example illustrating this intricate cross-kingdom communication involves the epigenetic regulation of the forkhead box P3 (FOXP3) transcription factor by gut microbiota-derived metabolites and signaling molecules (Figure 2).

### 5.1. Epigenetic Regulation of FOXP3 by the Gut Microbiome

The intestinal mucosal immune system requires a finely tuned balance between tolerance and defense to maintain homeostasis in the presence of a vast and dynamic microbial ecosystem [104]. Central to this regulation are regulatory T cells (T_reg_), which express the FOXP3 transcription factor and play critical roles in suppressing inflammatory responses, promoting tolerance to commensals, and preserving epithelial integrity [105,106].

The stable expression of FOXP3 is epigenetically governed through both DNA methylation and histone modifications. Specifically, demethylation of the conserved non-coding sequence 2 (CNS2) region within the FOXP3 locus is essential for transcriptional activation and heritable T_reg_ lineage stability [107]. This demethylation is facilitated by TET family enzymes (TET1–3), whose activity is enhanced by ascorbic acid (vitamin C), a cofactor that promotes chromatin accessibility and facilitates transcription factor binding [108].

Mouse models have demonstrated that T_reg_ differentiation and FOXP3 expression are profoundly shaped by the gut microbiota. GF or microbiota-depleted mice exhibit impaired T_reg_ development and abnormal immune activation. Notably, Foxp3-deficient mice display both elevated intestinal inflammation and altered gut microbial composition, including increased Firmicutes and reduced abundance of beneficial Clostridia and Lachnospiraceae [106,109,110]. Reintroduction of specific Clostridium strains restored the immunoregulatory microenvironment by inducing transforming growth factor-beta (TGF-β) production and facilitating FOXP3^+^ T_reg_ differentiation in the colon [110]. These findings are echoed in human immunodeficiency syndromes, such as IPEX (immune dysregulation, polyendocrinopathy, enteropathy, X-linked), where FOXP3 loss-of-function mutations lead to severe autoimmunity and disrupted microbiota homeostasis [111].

Among the most well-characterized microbial influences on FOXP3 regulation are microbial metabolites that act through epigenetic and immunomodulatory mechanisms. Butyrate, a short-chain fatty acid produced by *Clostridium* species, functions as an HDAC inhibitor, thereby enhancing histone acetylation at the FOXP3 promoter and CNS3 enhancer regions. This increase in acetylation fosters a more transcriptionally permissive chromatin state, facilitating sustained FOXP3 expression and stable T_reg_ differentiation [106,112]. In addition to SCFAs, secondary bile acid metabolites, such as isoalloLCA, have also been shown to stimulate FOXP3 transcription. This effect is mediated through the induction of mitochondrial reactive oxygen species (ROS), which can influence local chromatin accessibility and support the epigenetic stability of the T_reg_ lineage [113]. Another key microbial product is the capsular polysaccharide A (PSA) produced by *Bacteroides fragilis*, which enhances the function of FOXP3^+^ T_reg_ cells by upregulating CD39 and stimulating the secretion of interleukin-10 (IL-10) and interferon-gamma (IFN-γ), thereby reinforcing immunosuppressive pathways. In germ-free mice, the absence of PSA-producing strains results in diminished FOXP3^+^ T_reg_ populations and impaired mucosal tolerance—deficits that can be restored by colonization with PSA-expressing *B. fragilis* strains [72].

At the chromatin level, the interplay between DNA demethylation and histone acetylation establishes an open transcriptional environment. This epigenetic landscape allows bromodomain-containing (BRD) proteins to bind acetylated lysines and sustain FOXP3 expression. Pharmacological inhibition of BRD proteins using small molecules like JQ1 disrupts this interaction and has been proposed as a therapeutic strategy to modulate T_reg_ function in inflammatory and autoimmune diseases [105].

### 5.2. Broader Implications

The example of FOXP3 epigenetic regulation underscores a broader paradigm in which microbial-derived metabolites, surface polysaccharides, and molecular signals function as potent mediators of host–microbiome crosstalk, particularly within the immune compartment [105,106,107]. While FOXP3 remains among the most thoroughly studied targets of microbial modulation, emerging evidence suggests that a wide array of host regulatory genes and signaling pathways—including those involved in T helper cell differentiation, antigen presentation, and cytokine production—may be similarly influenced by microbiome-dependent epigenetic mechanisms [104].

For instance, butyrate and other short-chain fatty acids have been shown to modulate histone acetylation and methylation in dendritic cells and macrophages, altering their capacity to shape adaptive immune responses [68,79]. Similarly, microbial-derived indole derivatives [114], bile acid metabolites [113], and extracellular vesicles [86,87] can affect the expression of genes involved in barrier integrity and inflammation by altering local chromatin states.

These observations suggest that the microbiota contributes not only to immune education during development but also to dynamic epigenetic reprogramming in response to environmental cues throughout life. Understanding these broader interactions provides a conceptual framework for developing precision therapeutics aimed at restoring immunological tolerance or dampening pathological inflammation through microbiota-informed epigenetic targeting.

## 6. Future Research Strategies to Study the Microbiome–Epigenome Axis

Despite recent insights into how microbial communities modulate host DNA and RNA methylation through defined and evolutionarily conserved biochemical pathways, our ability to interrogate these processes at high resolution and translate them into clinically meaningful outcomes remains limited. Advancing this field will require the convergence of cutting-edge methylome profiling tools, longitudinal multi-omics studies, machine learning-based integrative models, and context-aware experimental systems. Moreover, aligning mechanistic discoveries with therapeutic development, biomarker validation, and equitable data governance will be critical for moving from bench to bedside. The following section outlines the most promising strategies and tools shaping this next phase of microbiome–epigenome research (Figure 3).

### 6.1. Mapping Methylation Landscapes with High-Throughput Technologies

Modern epigenomic research benefits from increasingly sensitive tools for profiling DNA and RNA methylation marks across diverse organisms. Early assumptions that many model organisms lacked significant methylation (e.g., *Drosophila melanogaster*) have been revised though the application of HPLC, LC-MS/MS [115], whole-genome bisulfite sequencing (WGBS), RRBS, and emerging single-molecule real-time (SMRT) sequencing platforms, which allow direct detection of methylated cytosine and adenine residues with ever higher sensitivity and resolution.

These technologies are now widely applied to bacteria, fungi, plants, and animals to uncover lineage-specific methylation patterns and discover novel methyltransferases [116,117]. When coupled with RNA-Seq, proteomics, and chromatin accessibility assays (ChIP-Seq, ATAC-Seq, etc.), they enable researchers to link methylation states with transcriptional activity, protein expression, and functional phenotypes. In microbial contexts, phase-variable restriction-modification systems, such as those characterized in *Helicobacter pylori* [117], *Campylobacter jejuni* [118], and *Neisseria* species [119], illustrate how epigenetic switches can drive rapid changes in gene expression, virulence, and immune evasion [120]. Resolving whether these microbial modifications can, in turn, reshape host epigenetic programs remains a compelling area for further study. Multi-omics techniques—including joint profiling of transcriptomics, methylomics, proteomics, and metabolomics in single cells [121], are becoming increasingly valuable for resolving cell-type-specific regulatory networks in response to microbial stimuli.

### 6.2. Systems-Level Integration Using Longitudinal Multi-Omics and AI

To fully elucidate microbiome–epigenome interactions, there is a growing need to move beyond static or reductionist approaches and toward dynamic, systems-level models. One promising direction involves the integration of high-dimensional multi-omics platforms across longitudinal cohorts. Studies such as the Survey of the Health of Wisconsin (SHOW) and the Qatar-based Omouma cohort have employed time-series analyses of host methylation, microbial taxa, and metabolic outputs to identify predictive biomarkers of disease susceptibility and progression [122,123]. Such designs can support causal inference and provide critical insight into temporal relationships between microbial community dynamics and host epigenetic states.

On the computational front, deep learning frameworks such as MethylNet [124] and DeepCpG [125] have shown promise in predicting cell-type-specific methylation patterns by integrating genomic, microbial, and environmental features. Meanwhile, spatial–temporal modeling tools like MEFISTO enable the integration of diverse data types (e.g., transcriptome, methylome, and metabolome) while preserving biological context [126]. While these tools will require greater interpretability and validation in diverse real-world populations, applying them in simplified experimental models may enable new insights into the mechanisms underlying microbial–epigenomic interactions.

### 6.3. Emerging Experimental Models to Study Microbial–Epigenomic Interactions

Despite methodological progress, dissecting the causal pathways that link microbial signaling molecules and the host epigenetic machinery is inherently challenging. Many studies rely on reductionist models that do not capture the full ecological complexity of gut microbial communities or the physiological state of host tissues. For example, germ-free or mono-colonized mice remain standard models but have limited translational value for recapitulating human–microorganism interactions [127]. Additionally, phase-variable restriction–modification systems and microbial methyltransferases can respond dynamically to environmental signals, horizontal gene transfers, and phage exposure, adding further complexity to their regulatory roles.

To address these limitations, innovative ex vivo and in vitro models are being developed. For instance, three-dimensional (3D) organoid cultures [128,129,130] and gut-on-a-chip [131,132,133] systems allow for co-culturing human IECs with microbial communities under controlled, physiologically relevant conditions. These systems can replicate nutrient gradients, oxygen level, and epithelial integrity, facilitating real-time monitoring of host–microorganism interactions [128]. The well-studied *Euprymna scolopes* (squid)—*Vibrio fischeri* (bacteria) symbiosis represents a tractable animal model for investigating how colonization can modulate host gene expression, given the interactions of a single bacterial species with its host on regulating squid development, health, and behavior [134,135,136]. Although DNA methylation has been documented in the squid [137], the epigenetic basis of this symbiosis remains underexplored and presents a highly simplified experimental system to more precisely explore the functional relationships between the endogenous host epigenetic processes and microbial symbiont factors that together shape host phenotypes.

### 6.4. From Mechanism to Translation: Clinical and Therapeutic Frontiers

A critical objective for the field is to translate mechanistic insights into actionable clinical interventions that harness microbiome–epigenome interactions. Live biotherapeutic products (LBPs), which include engineered or naturally occurring microbial strains capable of secreting beneficial metabolites such as butyrate, inosine, or PSA, are actively being evaluated in early-phase clinical trials for a range of immune-mediated diseases, including IBD, atopic dermatitis, and autoimmune arthritis [138]. These microbes exert immunomodulatory effects in part through epigenetic reprogramming of intestinal and immune cells, such as the induction of histone acetylation in regulatory T cells or suppression of DNA methyltransferase activity in macrophages [91].

Beyond targeted LBPs, broader microbiome-modulating strategies such as fecal microbiota transplantation (FMT) have gained prominence, particularly in the treatment of recurrent *Clostridioides difficile* infections [139,140,141]. However, their utility in treating extraintestinal conditions such as metabolic syndrome, autism spectrum disorders, and even cancer remains investigational. Studies suggest that FMT can induce durable shifts in host methylation profiles, including in genes involved in lipid metabolism, neurodevelopment, and inflammatory cascades—highlighting the need to decode the epigenetic consequences of engraftment and donor–recipient compatibility [142]. The long-term success of such interventions may depend on epigenomic “readiness” of host tissues to respond to microbial stimuli, a concept that remains poorly understood.

In oncology, the intersection between microbial metabolites and cancer epigenetics is beginning to draw attention. For instance, microbial production of SCFAs can modulate HDAC activity in tumor cells, potentially influencing cancer progression and response to epigenetic drugs [143]. Tumor-associated dysbiosis has also been linked to altered methylation patterns in colorectal and hepatocellular carcinomas, raising the possibility that microbiome-based interventions could re-sensitize tumors to epigenetic therapies [144].

To achieve safe and effective clinical translation, stratification of patients based on combined microbiome and epigenome signatures will be essential. Biomarkers of therapeutic responsiveness may include DNA methylation patterns in key regulatory genes, chromatin accessibility in immune cell populations, or metabolite signatures indicative of microbial activity. Initiatives like the PRISM consortium and the Human Microbiome Project Phase 2 (Integrative Human Microbiome Project) are beginning to integrate multi-omic data at scale to define such predictive profiles [145].

However, this progress must also contend with confounding biological and social variables. Factors such as age, biological sex, dietary habits, geographic location, and ancestry shape both microbial communities and epigenetic landscapes in non-linear ways [146]. Adaptive clinical trial designs, including basket trials and real-world registry-based studies, have been proposed as promising frameworks to accommodate biological heterogeneity in personalized medicine and may be adaptable to microbiome–epigenome research [147].

Ultimately, realizing the therapeutic potential of microbiome–epigenome research will require not only mechanistic precision, but also the development of delivery systems, dosing regimens, and regulatory frameworks that account for the dynamic and individualized nature of host epigenetic–microbial crosstalk.

### 6.5. Standards, Equity, and Open Data Stewardship

The reproducibility and translational impact of microbiome–epigenome research hinges on rigorous technical and ethical standardization. Key experimental variables—including reference DNA, spike-in controls, methylation quantification protocols, and harmonized procedures for biospecimen collection and stabilization—must be systematically benchmarked to minimize batch effects and ensure cross-cohort comparability. This is particularly crucial for global multi-center consortia where analytical variability can obscure genuine biological signals.

Several efforts are actively addressing these challenges. The International Human Epigenome Consortium (IHEC) [148] and the National Institute of Standards and Technology (NIST) [149] have released standardized protocols for bisulfite sequencing and quality control guidelines for epigenomic data generation, helping to reduce inter-laboratory discrepancies. Similarly, the Human Microbiome Project (HMP) [145] and MetaSUB (Metagenomics and Metadesign of the Subways and Urban Biomes) [150] have adopted platform-independent microbial metagenomic workflows that could be adapted for integrated epigenomic applications.

Beyond laboratory and analytical pipelines, robust metadata annotation and data stewardship practices are essential. The FAIR Data Principles—Findable, Accessible, Interoperable, and Reusable—have become foundational in microbiome and omics research, promoting transparency, reusability, and machine-readability of data [151]. Public platforms such as MG-RAST, EBI Metagenomics, and GEO (Gene Expression Omnibus) increasingly require detailed metadata about sample origin, processing conditions, and participant demographics, improving data quality for meta-analyses.

However, as microbiome–epigenome studies expand into diverse human populations, attention to equity and justice in data governance becomes equally essential. The CARE principles—Collective Benefit, Authority to Control, Responsibility, and Ethics—developed by the Global Indigenous Data Alliance (GIDA), emphasize that open science must also serve the interests of historically marginalized and sovereign indigenous communities [152]. These principles are not opposed to FAIR but serve as a necessary complement, especially in contexts where data extraction risks undermining community autonomy [153], which is particular important for Indigenous Peoples.

Recent examples of CARE implementation include the Silent Genomes Project in Canada, which developed governance models for genomic data in indigenous populations, and Te Mana Raraunga in Aotearoa New Zealand, which works to embed Māori values into national data infrastructures [154,155,156,157,158]. These frameworks provide vital blueprints for integrating ethical considerations into microbiome–epigenome consortia, particularly those involving ancestrally diverse or vulnerable populations.

## 7. Conclusions and Future Directions

The human gut microbiome represents a metabolically active and ecologically dynamic system that not only shapes, but is shaped by, the host’s epigenetic architecture. Mounting evidence now implicates gut microorganisms as modulators of host DNA and RNA methylation through multiple mechanistically distinct but converging routes—including nutrient provisioning for one-carbon metabolism, polyamine biosynthesis, SCFA production, and the release of extracellular vesicles loaded with regulatory molecules. In turn, host genomic and epigenomic contexts, including genetic variants, immune regulation, dietary inputs, and aging, exert selective pressures on microbial composition, metabolic output, and ecological stability.

Despite notable advances in deciphering these bidirectional relationships, many mechanistic and translational questions remain open. Key challenges include disentangling species- and tissue-specific effects, modeling microorganisms–host interactions under physiologically relevant conditions, and establishing causality in complex, polymicrobial settings. Reductionist models such as germ-free or mono-colonized mice, or even emerging models such as the squid-*Vibrio fischeri,* have yielded foundational insights; yet their translational relevance to human health remains limited. To bridge this gap, high-resolution platforms—including single-cell and spatial omics, 3D intestinal organoids, and gut-on-a-chip systems—are now enabling a more granular understanding of host-microbiota interactions within intact tissue contexts.

Simultaneously, the integration of longitudinal multi-omics studies with machine learning frameworks is opening new avenues for predictive modeling of epigenetic reprogramming and disease trajectories. These computational approaches, when paired with population-scale datasets, hold promise for identifying actionable biomarkers, therapeutic targets, and individualized intervention strategies. Interventions such as LBPs and FMT are already advancing into early-phase clinical trials for immune, metabolic, and neurological disorders, though their success will depend heavily on stratifying patients by microbial and epigenomic profiles. Innovative trial designs—including basket and registry-based studies—may be better suited to address population heterogeneity and improve translational fidelity.

In parallel, ensuring scientific rigor and equity in this emerging field will require robust standards for biospecimen handling, sequencing, metadata annotation, and data sharing. The adoption of FAIR principles has greatly enhanced reproducibility and data interoperability across microbiome and epigenome research. Complementary ethical frameworks such as the CARE principles are also essential to safeguard sovereignty and justice for underrepresented communities, especially in global and Indigenous genomics collaborations.

Ultimately, decoding the microbiome–epigenome interface offers unprecedented opportunities for precision medicine. This includes not only targeted epigenetic or microbial therapies, but also anticipatory approaches such as microbiome-informed nutrition and early disease risk stratification. Realizing this potential will demand sustained interdisciplinary collaboration across microbiology, genetics, epigenetics, computational biology, and clinical sciences, grounded in both methodological rigor and social responsibility.

## Figures and Tables

**Figure 1 ijms-26-08658-f001:**
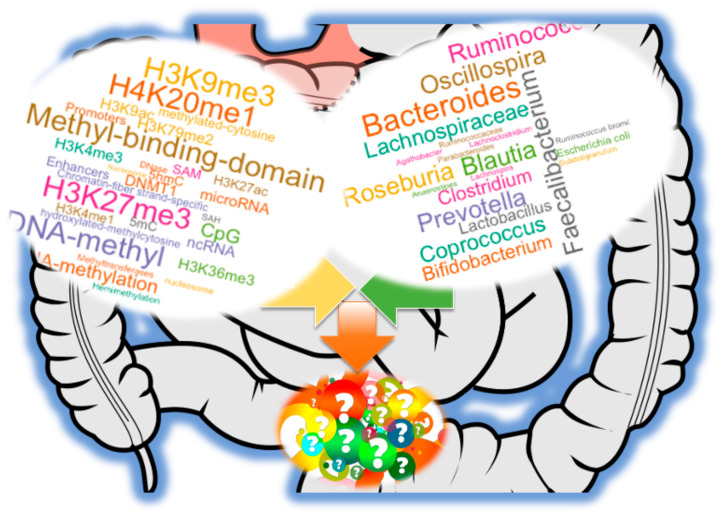
Gut microbiome influence host epigenomic regulation. The microbiome encompasses the entire community of microorganisms, their collective genetic material, and their interactions within a given environment. In contrast, the microbiota refers specifically to the living microorganisms themselves. The gut microbiome is closely linked to host epigenomic regulation and is increasingly recognized for its role in modulating DNA and RNA methylation. However, the precise molecular mechanisms by which these microorganisms direct host methylation remain incompletely understood. This figure highlights representative microbial genera commonly found in healthy human populations that have been reported to interact with components of host methylome regulation at the cellular level.

**Figure 2 ijms-26-08658-f002:**
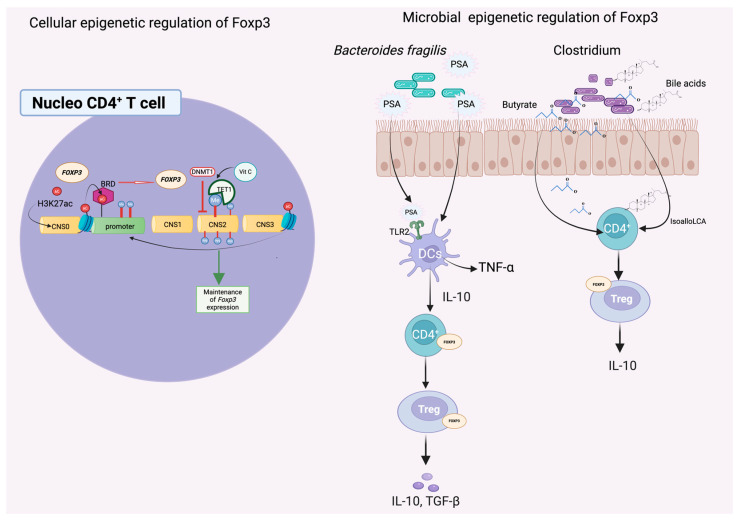
Epigenetic Regulation of FOXP3 Expression by Microbial Metabolites. The FOXP3 autoregulatory system functions as a critical negative regulator of immune responses and can be modulated by microbial-derived metabolites. In both human and mouse systems, sustained FOXP3 expression depends on the demethylation of CpG motifs within its promoter and the conserved non-coding sequence 2 (CNS2) region. Concurrently, active histone modifications establish an accessible chromatin environment that promotes the recruitment of transcriptional machinery and FOXP3 gene expression. Following induction, histone acetylation is facilitated by the binding of bromodomain-containing (BRD) proteins; this process is susceptible to inhibition by small molecules such as JQ1, which modulate FOXP3 transcriptional output. The DNA demethylase TET1, which requires ascorbic acid (vitamin C) as a cofactor, actively removes DNA methyltransferase 1 (DNMT1)-mediated repressive marks at the CNS2 region, thereby enhancing transcriptional activation. Several microbial metabolites—including capsular polysaccharide A (PSA) from *Bacteroides fragilis*, butyrate produced by *Clostridium* species, and bile acid derivatives—have been shown to trigger or support the activation of this FOXP3 regulatory network.

**Figure 3 ijms-26-08658-f003:**
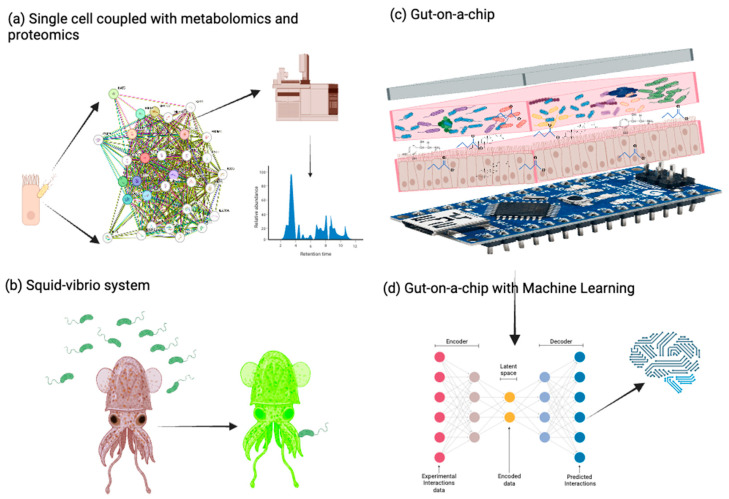
Experimental and computational strategies to investigate gut microbiome–intestinal epithelial cell interactions. This figure presents representative approaches for studying interactions between the gut microbiota and intestinal epithelial cells (IECs). (**a**) Single-cell analyses integrating metabolomic and proteomic data reveal cell-specific microbial effects and epigenomic responses. (**b**) The squid–Vibrio symbiosis serves as a tractable model system in which colonization of *Euprymna scolopes* by *Vibrio fischeri* induces bioluminescence, providing insights into microbe–host communication. (**c**) Advanced three-dimensional (3D) culture systems, including gut-on-a-chip models, enable co-culturing of IECs with microbial communities under physiologically relevant conditions. (**d**) Machine learning algorithms are increasingly used to integrate multi-omics data and predict microbiota–epigenome interactions.

**Table 1 ijms-26-08658-t001:** Cross-kingdom gene pathways of epigenetic methylation systems.

	System		DNA Methylation	DNA Demethylation	RNA Methylation	References
Bacterial	Restriction- Modification (RM)	4mC	N4-methylcytosine			[20,21]
		5mC	N5-Methylcytosine			
		6mA	N6-adenine methylation			
	Bacterial small RNA regulators (sRNAs)	C2 and C8 carbons of adenosines			*Ksg*A, *Rlm*A *Rsm*G	[22]
Fungal	Repeat-Induced Point mutation (RID)	5mC	Masc1, Masc2 DNMT1, DNMT5,			[23]
		6mA at ApT dinucleotides				[24,25]
					RNA Polymerase IV, Polymerase V RDR6, NERD	[26]
Plants	Methylation	CG context	Methyltransferase 1	DNA glycosylases demeter, and demeter-like 2 and 3		[27,28,29]
		CHGs	Chromomethylase 2 (CMT2)			
		CHH	methylase 2 or CMT2			
	RNA-direct DNA Methylation (RdRM)				RNA Polymerase IV, Polymerase V RDR6, NERD	[26,30]
Humans	Methylation	5mC	DNMT1, DNMT3a, DNMT3b; DNMT3L (noncatalytic but regulatory cofactor)	TET1, TET2, and TET3 5-hydroxymethylcytosine (5hmC), then to 5-formylcytosine (5foC), and ultimately to 5-carboxylcytosine (5caC)		[3,31,32]
		m6A			Writers: METTL3, METTL14, METTL16, WTAP	[33,34,35]
					Erasers: FTO, ALKBH5	
					Readers: YTHDF1-3 and YTHDC1-2, IGF2BP1-3	
	RNA methylation	tRNA^Asp^			DNMT2	[36,37]

**Table 2 ijms-26-08658-t002:** Microbial biochemical products and contribution to the host epigenome.

Process	Substrate	Enzyme	Product	Species	Epigenomic Contribution	References
Vitamins biosynthesis				Bacteroidetes Fusobacteria Proteobacteria		[61]
	Folate (Vit B9) 5-methyl-THF	Methionine Synthase (MS) + B12 + Homocysteine	Methionine + Tetrahydrofolate (THF)	*Lactobacilli*		[62]
				*Bifidobacteria*		[63]
One-carbon metabolism	Methionine	Methionine adenosyltransferase (MAT)	S-Adenosylmethionine (SAM)		SAM, universal methyl donor	
Polyamine biosynthesis	Arginine	Arginase (RocF) Arginine deiminase (ADI) anaerobic bacteria	Ornithine + Urea	*E. coli* *B. subtilis* *Pseudomonas aeruginosa* *Helicobacter pylori* *Lactobacillus* spp.	SAM and dcSAM balance	
	Ornithine	Ornithine decarboxylase (ODC)	putrescine	*Bacillus* *Halomonas* *Lactobacillus* spp.		[64]
				*Shewanella xiamenensis*		[65,66]
	Putrescina + dcSAM	Spermidine synthase	Spermidine + Methyltioadenosine (MTA)	*Enterococcus faecalis* *Streptococcus* *thermophilus* *Shewanella xiamenensis* *Propionibacterium freundenreichii*		
	Spermidine + dsSAM	Spermine synthase	Spermine + MTA	*Bacillus subtilis* *E. coli*		
Tricarboxylic acid cycle (TCA)	Succinate	Succinate dehydrogenase	Fumarate		Inhibits histone and DNA demethylases	[67,68]
	Glucosinolates	Myrosinase	Isothiocyanato + glucose + Sulfate	*Bacteroides thetaiotaomicron*		[69]
			sulfaraphane		Inhibits histone deacetylases and DNMTs	[70]
Extracellular Vesicles				*Pseudomona aeruginosa*	Reduces methylation levels in macrophages	[71]

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
