# Peer review of "The Gut Microbiome and Epigenomic Reprogramming: Mechanisms, Interactions, and Implications for Human Health and Disease"

_ijms, 2025, doi:10.3390/ijms26178658_

Round 1
Reviewer 1 Report
Comments and Suggestions for Authors
The first half, titled "3. Mechanistic insights into microbiota-induced host epigenetic regulation," is particularly noteworthy. It systematically describes the five principal routes through which gut bacteria alter host DNA and RNA methylation. These five routes are: one-carbon metabolism, polyamine metabolism, fermentation products, extracellular vesicles, and inflammatory signaling. The text is well-written and engaging, which makes it a valuable resource for those interested in this field. Conversely, the second half, particularly section 6, titled "Future areas of promising research strategies to study these interactions," is comparatively superficial given the scope and variety of challenges facing the field. It also omits several critical perspectives.
First, the text lacks concrete research detail. Beyond a general overview of multi-omics and AI, the text would be more convincing if it summarized specific, ongoing studies.
Second, virtually no bridge has been built to clinical translation. Issues such as trial design, GMP-grade microbial therapeutics, biomarker discovery frameworks, and how to accommodate cohort diversity (age, ethnicity, diet) should be addressed.
Thirdly, standardization is scarcely mentioned. Establishing cross-site sample-processing protocols, methylation benchmarks (reference DNA, spike-in controls), and guidelines for depositing metadata in public databases are all platform elements that will shape future work.
Further exploration of these topics would enhance the review's relevance for researchers and clinicians alike.
Author Response
Comment 1:
The first half, titled "3. Mechanistic insights into microbiota-induced host epigenetic regulation," is particularly noteworthy. It systematically describes the five principal routes through which gut bacteria alter host DNA and RNA methylation. These five routes are: one-carbon metabolism, polyamine metabolism, fermentation products, extracellular vesicles, and inflammatory signaling. The text is well-written and engaging, which makes it a valuable resource for those interested in this field.
Conversely, the second half, particularly section 6, titled "Future areas of promising research strategies to study these interactions," is comparatively superficial given the scope and variety of challenges facing the field. It also omits several critical perspectives:
- First, the text lacks concrete research detail. Beyond a general overview of multi-omics and AI, the text would be more convincing if it summarized specific, ongoing studies.
- Second, virtually no bridge has been built to clinical translation. Issues such as trial design, GMP-grade microbial therapeutics, biomarker discovery frameworks, and how to accommodate cohort diversity (age, ethnicity, diet) should be addressed.
- Thirdly, standardization is scarcely mentioned. Establishing cross-site sample-processing protocols, methylation benchmarks (reference DNA, spike-in controls), and guidelines for depositing metadata in public databases are all platform elements that will shape future work.
Further exploration of these topics would enhance the review's relevance for researchers and clinicians alike.
Response 1:
We thank the reviewer for this valuable and constructive feedback. In response, we have substantially revised and expanded Section 6 to address these concerns and enhance the utility of the review for both basic and translational researchers.
Specifically, we have:
- Reorganized and retitled Section 6 to provide a clearer conceptual structure, integrating mechanistic, technological, translational, and ethical perspectives. It now includes five thematically coherent sub-sections (6.1–6.5), each with updated literature and examples.
- Included specific, ongoing multi-omics studies and platforms, such as the Integrative Human Microbiome Project (iHMP) and MetaSUB, which exemplify current efforts to resolve microbiota–epigenome interactions in human populations and diverse environments.
- Integrated emerging computational frameworks, including DeepCpG, MethylNet, and other AI-driven tools that facilitate multi-layer data integration and methylation prediction in heterogeneous tissue or microbial samples.
- Expanded the translational outlook, covering early-phase trials involving live biotherapeutic products (LBPs), the use of fecal microbiota transplantation (FMT) in epigenome-sensitive contexts, and the development of microbiome- and methylation-based biomarkers for disease risk prediction. We also describe the potential of adaptive trial designs (e.g., basket and registry-based trials) to accommodate cohort diversity (age, sex, diet, and ancestry) and improve clinical relevance.
- Added a new subsection on standardization and open data stewardship, emphasizing the importance of reference materials (e.g., spike-in controls), harmonized biospecimen handling protocols, and metadata annotation. This includes discussion of both the FAIR (Findable, Accessible, Interoperable, Reusable) and CARE (Collective Benefit, Authority to Control, Responsibility, Ethics) data principles, to highlight the importance of both technical reproducibility and ethical inclusivity in global consortia and Indigenous data governance.
We believe these additions significantly strengthen the manuscript’s scope, methodological depth, and translational applicability. The revised Section 6 now appears on lines [540-700] of the manuscript. We thank the reviewer again for prompting these meaningful improvements.
Reviewer 2 Report
Comments and Suggestions for Authors
I have carefully reviewed the article entitled "The Gut Microbiome and Epigenomic Reprogramming: Mechanisms, Interactions, and Implications for Human Health and Disease" and I have found some major flaws, which need to be revised before further processing.
Abstract
The term mentioned in the title “Gut Microbiome” and “gut microbiota” in first line of abstract and so on… may need check for similarity, although the meaning of both is almost the same.
Line 13. Texts “dynamic, complex, and diverse community” have almost the same sense, please revise the sentence with different texts and sense. Although we know a lot about their functions, so please revise the whole sentence. Or mentioned “…the underlying molecular mechanisms…. are not yet fully understood…”.
Line 18-19. …influenced by environmental factors, including microbial signals. Please check and revise as environmental factors, and microbial signals are two different things.
Line 19-20. Please check this sentence and compared with the first line, make sure that the line may need be repeated in the abstract.
Line 21. Please start with “This review overview or highlight…the word synthesizes is not a suitable here.
Line 25-26, and line 71. A review article cannot explore…, please replace the “explore” with another word.
Keywords: Microbiome, either use the word “microbiome” or “microbiota” in the article, starting from the title….
Overall paragraphs are too small, and the authors quickly jumped from one topic differ from the other.
It’s surprising to see the article is related to human and microbiota interactions, but the authors explaining 2.2. Yeast and fungal DNA methylation, 2.3. Plant DNA methylation, 2.4. DNA methylation and RNA regulatory activity in animals, etc. please carefully revised and restructure the review before resubmitting. The current review has major flaws to consider for publication. Please carefully revise the whole review article.
Author Response
Comment 1:
The term mentioned in the title “Gut Microbiome” and “gut microbiota” in first line of abstract and so on… may need check for similarity, although the meaning of both is almost the same.
Response 1:
We thank the reviewer for noting this inconsistency. While “microbiome” and “microbiota” are often used interchangeably, we have revised the manuscript to use "microbiome" or “microbiota” consistently in both the title, abstract, and body for terminological clarity. Note the following: the microbiome is the community of microorganisms and their genetic material that inhabit a specific environment. In humans, it can refer to the collective microbes residing in the entire body or in a particular site, such as the gut, skin, mouth, or respiratory tract. These microbial communities interact with the host and play important roles in health and disease. The microbiota refers to the living microorganisms themselves—such as bacteria, viruses, fungi, and archaea—that inhabit a specific environment or part of the body (e.g., gut microbiota, skin microbiota). It focuses on the actual organisms present, without including their collective genetic material or functional interactions. We further clarify this by defining these two terms in the figure legend of Figure 1, as it illustrates the overall focus of this review.
Comment 2:
Line 13. Texts “dynamic, complex, and diverse community” have almost the same sense, please revise the sentence with different texts and sense. Although we know a lot about their functions, so please revise the whole sentence. Or mentioned “…the underlying molecular mechanisms…. are not yet fully understood…”.
Response 2:
We agree and have revised the sentence to avoid redundancy and reflect a clearer focus on underlying mechanisms. We now have significantly revised the abstract, which addresses other concerns below.
Comment 3:
Line 18-19. …influenced by environmental factors, including microbial signals. Please check and revise as environmental factors, and microbial signals are two different things.
Response 3:
We thank the reviewer and have clarified this distinction in the revised the abstract.
Comment 4:
Line 19-20. Please check this sentence and compared with the first line, make sure that the line may need be repeated in the abstract.
Response 4:
We reviewed the section and found some conceptual overlap. We have revised to avoid redundancy while maintaining clarity. See the newly rewritten abstract.
Comment 5:
Line 21. Please start with “This review overview or highlight…the word synthesizes is not a suitable here.
Response 5:
We have replaced “synthesizes” with more appropriate academic phrasing (see new Abstract).
Comment 6:
Line 25-26, and line 71. A review article cannot explore…, please replace the “explore” with another word.
Response 6:
We agree and have replaced “explore” with more accurate, non-experimental language, such as “examine” or “highlight”.
Comment 7:
Keywords: Microbiome, either use the word “microbiome” or “microbiota” in the article, starting from the title….
Response 7:
We have revised the keywords to align with the terminology used in the title and text, just as mentioned above in a prior comment.
Comment 8:
Overall paragraphs are too small, and the authors quickly jumped from one topic differ from the other.
Response 8:
We acknowledge this issue and have significantly modified this by including expanded sections, including revised Section 6 (as well as to respond to Reviewer 1); and have substantially revised the entire review (see below).
Comment 9:
It’s surprising to see the article is related to human and microbiota interactions, but the authors explaining 2.2. Yeast and fungal DNA methylation, 2.3. Plant DNA methylation, 2.4. DNA methylation and RNA regulatory activity in animals, etc. please carefully revised and restructure the review before resubmitting. The current review has major flaws to consider for publication. Please carefully revise the whole review article.
Response 9:
We appreciate this feedback and agree that the relevance of Sections 2.2–2.4 must be clearer. We have revised the section introductions to clarify that these comparisons establish key epigenetic mechanisms across taxa before focusing specifically on human and gut microbiome contexts. You will see that we have substantially revised the entire review article to include more content relevant to the topic along with appropriate, recent examples, and have thoroughly proofread the entire text and, where necessary, improved its clarity, uniformity, and grammatical structure to eliminate any major flaws that would otherwise prevent publication.
Round 2
Reviewer 1 Report
Comments and Suggestions for Authors I am in agreement with the acceptance of this paper as the authors have adequately addressed the concerns of my peer review.Reviewer 2 Report
Comments and Suggestions for Authors
Despite our explicit request that the author remove the sections about plants and other organisms from the manuscript, which go against the title and the stated focus on human-related study, the author has added more irrelevant text instead of addressing the issue. We must unfortunately, reject the article because the manuscript still contains information that is outside the purview of the study and contradicts its stated goals. In summary, the author has added more irrelevant text rather than eliminating the sections that were deemed unnecessary as requested. Further, the writing/structure of the review shows a lack of knowledge about how to write a scholarly article with a clear structure.